# Schur-Convexity of the Mean of Convex Functions for Two Variables

**Huan-Nan Shi [1], Dong-Sheng Wang [2] and Chun-Ru Fu [3,\*]**

1 Department of Electronic Information, Teacher's College, Beijing Union University, Beijing 100011, China
2 Basic Courses Department, Beijing Polytechnic, Beijing 100176, China
3 Applied College of Science and Technology, Beijing Union University, Beijing 102200, China
\* Correspondence: fuchunru2008@163.com

**Abstract:** The results of Schur convexity established by Elezovic and Pecaric for the average of convex functions are generalized relative to the case of the means for two-variable convex functions. As an application, some binary mean inequalities are given.

**Keywords:** inequality; Schur-convex function; Hadamard's inequality; convex functions of two variables; mean

**MSC:** 26A51; 26D15; B25

## 1. Introduction

Let $\mathbb{R}$ be a set of real numbers, $g$ be a convex function defined on the interval $I \subseteq \mathbb{R} \to \mathbb{R}$ and $c, d \in I$, $c < d$. Then

$$g\left(\frac{d+c}{2}\right) \leq \frac{1}{d-c}\int_c^d g(t)\,dt \leq \frac{g(d)+g(c)}{2}. \tag{1}$$

This is the famous Hadamard's inequality for convex functions.

In 2000, utilizing Hadamard's inequality, Elezovic and Pecaric [1] researched Schur-convexity on the lower and upper limit of the integral for the mean of the convex functions and obtained the following important and profound theorem.

**Theorem 1** ([1]). *Let $I$ be an interval with nonempty interior on $\mathbb{R}$ and $g$ be a continuous function on $I$. Then,*

$$\Phi(c,d) = \begin{cases} \frac{1}{d-c}\int_c^d g(s)ds, & c,d \in I, \ d \neq c \\ g(c), & d = c \end{cases}$$

*is Schur convex (Schur concave, resp.) on $I \times I$ iff $g$ is convex (concave, resp.) on $I$.*

In recent years, this result attracted the attention of many scholars (see references [2–12] and Chapter II of the monograph [13] and its references).

In this paper, the result of theorem 1 is generalized to the case of bivariate convex functions, and some bivariate mean inequalities are established.

**Theorem 2.** *Let $I$ be an interval with non-empty interior on $\mathbb{R}$ and $g(s,t)$ be a continuous function on $I \times I$. If $g$ is convex (or concave resp.) on $I \times I$, then*

$$G(u,v) = \begin{cases} \frac{1}{(v-u)^2}\int_u^v \int_u^v g(s,t)\,ds\,dt, & (u,v) \in I \times I, \ u \neq v \\ g(u,u), & (u,v) \in I \times I, \ u = v \end{cases} \tag{2}$$

*is Schur convex (or Schur concave, resp.) on $I \times I$.*

## 2. Definitions and Lemmas

To prove Theorem 2, we provide the following lemmas and definitions.

**Definition 1.** *Let $(x_1, x_2)$ and $(y_1, y_2) \in \mathbb{R} \times \mathbb{R}$.*

*(1)   A set $\Omega \subset \mathbb{R} \times \mathbb{R}$ is said to be convex if $(x_1, x_2), (y_1, y_2) \in \Omega$ and $0 \leq \beta \leq 1$ implies*

$$(\beta x_1 + (1 - \beta)y_1, \beta x_2 + (1 - \beta)y_2) \in \Omega.$$

*(2)   Let $\Omega \subset \mathbb{R} \times \mathbb{R}$ be convex set. A function $\psi \colon \Omega \to \mathbb{R}$ is said to be a convex function on $\Omega$ if, for all $\beta \in [0, 1]$ and all $(x_1, x_2), (y_1, y_2) \in \Omega$, inequality*

$$\psi(\beta x_1 + (1 - \beta)y_1, \beta x_2 + (1 - \beta)y_2) \leq \beta \psi(x_1, x_2) + (1 - \beta)\psi(y_1, y_2) \tag{3}$$

*holds. If, for all $\beta \in [0, 1]$ and all $(x_1, x_2), (y_1, y_2) \in \Omega$, the strict inequality in (3) holds, then $\psi$ is said to be strictly convex. $\psi$ is called concave ( or strictly concave, resp.) iff $-\psi$ is convex ( or strictly convex, resp.)*

**Definition 2** ([14,15])**.** *Let $\Omega \subseteq \mathbb{R} \times \mathbb{R}$, $(x_1, x_2)$ and $(y_1, y_2) \in \Omega$, and let $\varphi : \Omega \to \mathbb{R}$:*

*(1)   $(x_1, x_2)$ is said to be majorized by $(y_1, y_2)$ (in symbols $(x_1, x_2) \prec (y_1, y_2)$) if $\max\{x_1, x_2\} \leq \max\{y_1, y_2\}$ and $x_1 + x_2 = y_1 + y_2$.*

*(2)   $\psi$ is said to be a Schur-convex function on $\Omega$ if $(x_1, x_2) \prec (y_1, y_2)$ on $\Omega$ implies $\psi(x_1, x_2) \prec \psi(y_1, y_2)$, and $\psi$ is said to be a Schur-concave function on $\Omega$ iff $-\psi$ is a Schur-convex function.*

**Lemma 1** ([14] (p. 5))**.** *Let $(x_1, x_2) \in \mathbb{R} \times \mathbb{R}$. Then*

$$\left( \frac{x_1 + x_2}{2}, \frac{x_1 + x_2}{2} \right) \prec (x_1, x_2).$$

**Lemma 2** ([14] (p. 5))**.** *Let $\Omega \subseteq \mathbb{R} \times \mathbb{R}$ be symmetric set with a nonempty interior $\Omega^\circ$. $\psi : \Omega \to \mathbb{R}$ is continuous on $\Omega$ and differentiable in $\Omega^\circ$. Then, function $\psi$ is Schur convex (or Schur concave, resp.) iff $\psi$ is symmetric on $\Omega$ and*

$$(x_1 - x_2)\left( \frac{\partial \psi}{\partial x_1} - \frac{\partial \psi}{\partial x_2} \right) \geq 0 (or \leq 0, resp.)$$

*holds for any $(x_1, x_2) \in \Omega^\circ$.*

**Lemma 3** ([16])**.** *Let $\varphi(x, w)$ and $\frac{\partial \varphi(x, w)}{\partial w}$ be continuous on*

$$D = \{(x, w) : a \leq x \leq b, c \leq w \leq d\}; let$$

*$a(w), b(w)$ and their derivatives be continuous on $[c, d]$; $v \in [c, d]$ implies $a(w), b(w) \in [a, b]$. Then,*

$$\frac{\mathrm{d}}{\mathrm{d}w} \int_{a(w)}^{b(w)} \varphi(x, w)\, \mathrm{d}x = \int_{a(w)}^{b(w)} \frac{\partial \varphi(x, w)}{\partial w}\, \mathrm{d}x + \varphi(b(w), u)b'(w) - \varphi(a(w), w)a'(w). \tag{4}$$

**Lemma 4.** *Let $g(s, t)$ be continuous on rectangle $[a, p; a, q]$, $G(c, d) = \int_c^d \int_c^d g(s, t)\, \mathrm{d}s\, \mathrm{d}t$. If $c = c(b)$ and $d = d(b)$ are differentiable with $b$, $a \leq c(b) \leq p$ and $a \leq d(b) \leq q$, then*

$$\frac{\partial G}{\partial b} = \int_c^d g(s, d)d'(b)\, \mathrm{d}s - \int_c^d g(s, c)c'(b)\, \mathrm{d}s$$

$$+ d'(b) \int_c^d g(d, t)\, \mathrm{d}t - c'(b) \int_c^d g(c, t)\, \mathrm{d}t. \tag{5}$$

**Proof.** Let $\varphi(s,b) = \int_c^d g(s,t)\,\mathrm{d}t$. Then,

$$\frac{\partial \varphi(s,b)}{\partial b} = g(s,d)d'(b) - g(s,c)c'(b).$$

By Lemma 3, we have

$$\begin{aligned}
\frac{\partial G}{\partial b} &= \frac{\mathrm{d}}{\mathrm{d}b} \int_c^d \varphi(s,b)\,\mathrm{d}s \\
&= \int_c^d \frac{\partial \varphi(s,b)}{\partial b}\,\mathrm{d}s + \varphi(d,b)d'(b) - \varphi(c,b)c'(b) \\
&= \int_c^d g(s,d)d'(b)\,\mathrm{d}s - \int_c^d g(s,c)c'(b)\,\mathrm{d}s \\
&\quad + d'(b) \int_c^d g(d,s)\,\mathrm{d}s - c'(b) \int_c^d g(c,s)\,\mathrm{d}s.
\end{aligned}$$

$\square$

**Remark 1.** *In passing, it is pointed out that (9) in Lemma 5 of reference [2] is incorrect and should be replaced by (4) of this paper.*

**Lemma 5.** *Let $I$ be an interval with nonempty interior on $\mathbb{R}$ and $g(s,t)$ be a continuous function on $I \times I$. For $(u,v) \in I \times I, u \neq v$, let $G(u,v) = \int_u^v \int_u^v g(s,t)\,\mathrm{d}s\,\mathrm{d}t$. Then,*

$$\frac{\partial G}{\partial v} = \int_u^v g(s,v)\,\mathrm{d}s + \int_u^v g(v,t)\,\mathrm{d}t, \tag{6}$$

$$\frac{\partial G}{\partial u} = -\left( \int_u^v g(s,u)\,\mathrm{d}s + \int_u^v g(u,t)\,\mathrm{d}t \right). \tag{7}$$

**Proof.** By taking $c(b) = a$ and $d(b) = b$, we have $c'(b) = 0$ and $d'(b) = 1$. By (5) in Lemma 4, we obtain (6).

Notice that $G(u,v) = \int_v^u \int_v^u g(s,t)\,\mathrm{d}s\,\mathrm{d}t$; from (5), we have

$$\frac{\partial G}{\partial u} = \int_v^u g(s,u)\,\mathrm{d}s + \int_v^u g(u,t)\,\mathrm{d}t = -\left( \int_u^v g(s,u)\,\mathrm{d}s + \int_u^v g(u,t)\,\mathrm{d}t \right).$$

$\square$

**Lemma 6** ([14] (p. 38, Proposition 4.3) and [15] (p. 644, B.3.d)). *Let $\Omega \subset \mathbb{R} \times \mathbb{R}$ be an open convex set and let $\psi(x,y): \Omega \to \mathbb{R}$ be twice differentiable. Then, $\psi$ is convex on $\Omega$ iff the Hessian matrix*

$$H(x,y) = \begin{pmatrix} \frac{\partial^2 \psi}{\partial x \partial x} & \frac{\partial^2 \psi}{\partial x \partial y} \\ \frac{\partial^2 \psi}{\partial y \partial x} & \frac{\partial^2 \psi}{\partial y \partial y} \end{pmatrix}$$

*is non-negative definite on $\Omega$. If $H(x)$ is positive definite on $\Omega$, then $\psi$ is strictly convex on $\Omega$.*

## 3. Proofs of Main Results

**Proof of Theorem 2.** Let $g(s,t)$ be convex on $I \times I$. $G(u,v)$ is evidently symmetric. By Lemma 5, we have

$$\frac{\partial G(u,v)}{\partial v} = \frac{-2}{(v-u)^3} \int_u^v \int_u^v g(s,t)\,\mathrm{d}s\,\mathrm{d}t + \frac{1}{(v-u)^2} \left( \int_u^v g(s,v)\,\mathrm{d}s + \int_u^v g(v,t)\,\mathrm{d}t \right).$$

$$\frac{\partial G(u,v)}{\partial u} = \frac{2}{(v-u)^3} \int_u^v \int_u^v g(s,t)\,\mathrm{d}s\,\mathrm{d}t - \frac{1}{(v-u)^2} \left( \int_u^v g(s,u)\,\mathrm{d}s + \int_u^v g(u,t)\,\mathrm{d}t \right).$$

$$\Delta := (v - u)\left(\frac{\partial G(u,v)}{\partial v} - \frac{\partial G(u,v)}{\partial u}\right) = -\frac{4}{(v-u)^2}\int_u^v\int_u^v g(s,t)\,\mathrm{d}s\,\mathrm{d}t$$
$$+ \frac{1}{v-u}\int_u^v (g(s,v)+g(s,u))\,\mathrm{d}s + \frac{1}{v-u}\int_u^v (g(u,t)+g(v,t))\,\mathrm{d}t$$

By Hadamards inequality, we have

$$\frac{2}{(v-u)^2}\int_u^v\int_u^v g(s,t)\,\mathrm{d}s\,\mathrm{d}t = \frac{2}{v-u}\int_u^v\left(\frac{1}{v-u}\int_u^v g(s,t)\,\mathrm{d}s\right)\mathrm{d}t$$
$$\leq \frac{2}{v-ua}\int_u^v \frac{g(u,t)+g(v,t)}{2}\,\mathrm{d}t = \frac{1}{v-u}\int_u^v a(g(u,t)+g(v,t))\,\mathrm{d}t$$

and

$$\frac{2}{(v-u)^2}\int_u^v\int_u^v g(s,t)\,\mathrm{d}s\,\mathrm{d}t = \frac{2}{v-u}\int_u^v\left(\frac{1}{v-u}\int_u^v g(s,t)\,\mathrm{d}t\right)\mathrm{d}s$$
$$\leq \frac{2}{v-u}\int_u^v \frac{g(s,u)+g(s,v)}{2}\,\mathrm{d}s = \frac{1}{v-u}\int_u^v (g(s,u)+g(s,v))\,\mathrm{d}s.$$

Moreover, we have

$$\frac{4}{(v-u)^2}\int_u^v\int_u^v g(s,t)\,\mathrm{d}s\,\mathrm{d}t$$
$$\leq \frac{1}{v-u}\int_u^v (g(s,v)+g(s,u))\,\mathrm{d}s + \frac{1}{v-u}\int_u^v (g(u,t)+g(v,t))\,\mathrm{d}t.$$

Therefore, $\Delta \geq 0$, so $G(u,v)$ is Schur-convex on $I \times I$.

When $g(s,t)$ is a concave function on $I \times I$, it can be proved with similar methods. □

## 4. Application on Binary Mean

**Theorem 3.** *Let $c > 0$ and $d > 0$. If $c \neq d, 0 < s < 1$, then*

$$A(d,c) \geq S_{s+1}^s(d,c)S_s^{s-1}(d,c) \geq \frac{(c+d)^{2s-1}}{s(s+1)}, \tag{8}$$

*where $A(d,c) = \frac{c+d}{2}$ and $S_s(d,c) = \left(\frac{d^s-c^s}{s(d-c)}\right)^{\frac{1}{s-1}}$ are the arithmetic mean and the s-order Stolarsky mean of positive numbers c and d, respectively.*

**Proof.** Let $x > 0, y > 0$ and $0 < s < 1$. From Theorem 4 in the reference [17], we know that $g(x,y) = x^s y^{1-s}$ is concave on $(0,+\infty) \times (0,+\infty)$. For $c \neq d$, by Theorem 2, from $\left(\frac{d+c}{2},\frac{d+c}{2}\right) \prec (c,d) \prec (d+c,0)$, it follows that

$$G(d+c,0) = \frac{1}{(d+c-0)^2}\int_c^d\int_0^{d+c} x^s y^{1-s}\,\mathrm{d}x\,\mathrm{d}y$$
$$= \frac{1}{(d+c)^2}\int_0^{d+c} x^s\,\mathrm{d}x\int_0^{d+c} y^{1-s}\,\mathrm{d}y$$
$$= \frac{1}{(d+c)^2}\frac{(c+d)^{s+1}}{s+1}\frac{(c+d)^s}{s} = \frac{(c+d)^{2s-1}}{s(s+1)}$$
$$\leq G(c,d) = \frac{1}{(d-c)^2}\int_c^d\int_c^d x^s y^{1-s}\,\mathrm{d}x\,\mathrm{d}y$$
$$= \frac{1}{(d-c)^2}\int_c^d x^s\,\mathrm{d}x\int_c^d y^{1-s}\,\mathrm{d}y$$
$$= \frac{1}{(d-c)^2}\frac{d^{s+1}-c^{s+1}}{s+1}\frac{d^s-c^s}{s}$$
$$\leq G\left(\frac{d+c}{2},\frac{d+c}{2}\right) = \frac{d+c}{2},$$

That is, we obtain the following.

$$\frac{(c+d)^{2s-1}}{s(s+1)} \leq S_{s+1}^s(d,c)S_s^{s-1}(d,c) = \frac{d^{s+1}-c^{s+1}}{(s+1)(d-c)} \cdot \frac{d^s-c^s}{s(d-c)} \leq \frac{d+c}{2} = A(d,c).$$

□

**Theorem 4.** *Let $c > 0, d > 0$. Then,*

$$\log\left(\frac{A(d,c)}{B(d,c)}\right)^2 \geq \left(\frac{c-d}{d+c}\right)^2, \tag{9}$$

*where $B(d,c) = \sqrt{dc}$ is the geometric mean of of positive numbers c and d.*

**Proof.** From reference [17], we know that the function $g(x,y) = \frac{1}{(x+y)^2}$ is convex on $(0,+\infty) \times (0,+\infty)$. For $c > 0, d > 0$ and $d \neq c$, by Theorem 2, from $(\frac{d+c}{2}, \frac{d+c}{2}) \prec (d,c)$, it follows that

$$\begin{aligned}
G(c,d) &= \frac{1}{(d-c)^2} \int_c^d \int_c^d \frac{1}{(x+y)^2} \, dx \, dy \\
&= \frac{1}{(d-c)^2} \int_c^d \left(\frac{1}{c+y} - \frac{1}{d+y}\right) dy \\
&= \frac{1}{(d-c)^2} [(\log(d+c) - \log(2c)) - (\log(2d) - \log(d+c))] \\
&\geq G\left(\frac{d+c}{2}, \frac{d+c}{2}\right) = \frac{1}{(d+c)^2},
\end{aligned}$$

That is, we obtain the following.

$$\log\left(\frac{A(d,c)}{B(d,c)}\right)^2 = \log\frac{(d+c)^2}{4dc} \geq \left(\frac{c-d}{d+c}\right)^2.$$

□

**Theorem 5.** *Let $c > 0, d > 0$. Then,*

$$H_e(c^2, d^2) \geq A^2(c,d), \tag{10}$$

*where $H_e(c,d) = \frac{c+\sqrt{cd}+d}{3}$ is the Heronian mean of positive numbers c and d.*

**Proof.** From reference [18], we know that the function of two variables

$$\psi(x,y) = \frac{x^2}{2r^2} + \frac{y^2}{2s^2}$$

is a convex function on $(0,+\infty) \times (0,+\infty)$, where $s > 0$ and $r > 0$. For $d > 0, c > 0$, and $c \neq d$, by Theorem 2, from $(\frac{d+c}{2}, \frac{d+c}{2}) \prec (d,c)$, it follows that

$$
\begin{aligned}
G(c,d) &= \frac{1}{(d-c)^2} \int_c^d \int_c^d \left( \frac{x^2}{2r^2} + \frac{y^2}{2s^2} \right) \mathrm{d}x\,\mathrm{d}y \\
&= \frac{1}{(d-c)^2} \int_c^d \left( \frac{d^3-c^3}{6r^2} + \frac{y^2(d-c)}{2s^2} \right) \mathrm{d}y \\
&= \frac{1}{(d-c)^2} \left( \frac{(d^3-c^3)(d-c)}{6r^2} + \frac{(d^3-c^3)(d-c)}{6s^2} \right) \\
&= \frac{1}{(d-c)^2} \cdot \frac{(d^3-c^3)(d-c)}{6} \left( \frac{1}{r^2} + \frac{1}{s^2} \right) \\
&\geq G\left( \frac{d+c}{2}, \frac{d+c}{2} \right) = \frac{(c+d)^2}{8} \left( \frac{1}{r^2} + \frac{1}{s^2} \right),
\end{aligned}
$$

namely

$$
H_e(c^2, d^2) = \frac{c^2 + cd + d^2}{3} = \frac{(d^3 - c^3)}{3(d-c)} \geq \frac{(d+c)^2}{4} = A^2(d,c).
$$

□

**Theorem 6.** *Let $c > 0, d > 0$. We have*

$$
H_e(c^2, d^2) \geq L(d,c) A(d,c), \tag{11}
$$

*where $L(d,c) = \frac{d-c}{\log d - \log c}$ is the logarithmic mean of positive numbers $c$ and $d$.*

**Proof.** Let $g(x,y) = y^2 x^{-1}, x > 0, y > 0$. Then,

$$
g_{xx} = 2x^{-3}y^2, \;\; g_{xy} = -2x^{-2}y = g_{yx}, g_{yy} = 2x^{-1}.
$$

The Hesse matrix of $g(x,y)$ is

$$
H = \begin{pmatrix} 2x^{-3}y^2 & -2x^{-2}y \\ -2x^{-2}y & 2x^{-1} \end{pmatrix}.
$$

$$
\det(H - \lambda I) = \det \begin{pmatrix} 2x^{-3}y^2 - \lambda & -2x^{-2}y \\ -2x^{-2}y & 2x^{-1} - \lambda \end{pmatrix} = 0
$$

$$
\Rightarrow \lambda(\lambda - 2x^{-3}y^2 - 2x^{-1}) = 0 \Rightarrow \lambda_1 = 0, \lambda_2 = 2x^{-3}y^2 + 2x^{-1} > 0.
$$

Therefore, matrix $H$ is positive semidefinite, so it is known that $g(x,y)$ is a convex function on $(0,+\infty) \times (0,+\infty)$. For $d > 0, c > 0$ and $d \neq c$, by Theorem 2, from $\left( \frac{d+c}{2}, \frac{d+c}{2} \right) \prec (d,c)$, it follows that

$$
\begin{aligned}
G(c,d) &= \frac{1}{(d-c)^2} \int_c^d \int_c^d y^2 x^{-1} \,\mathrm{d}x\,\mathrm{d}y \\
&= \frac{\log d - \log c}{d - c} \cdot \frac{d^2 + cd + c^2}{3} \geq \frac{\left( \frac{d+c}{2} \right)^2}{\frac{c+c}{2}} = \frac{d+c}{2},
\end{aligned}
$$

which is

$$
H_e(c^2, d^2) \geq L(d,c) A(d,c).
$$

□

**Theorem 7.** *Let $d > 0, c > 0, d \neq c$. Then*

$$\widetilde{E}(d,c) \leq A(d,c)e^{(d+c)}\left(\frac{d-c}{e^d - e^c}\right)^2 \leq A(d,c), \tag{12}$$

*where*

$$\widetilde{E}(d,c) = \begin{cases} \frac{ce^d - de^c}{e^d - e^c} + 1, & d, c \in I, \ d \neq c \\ c, & c = d \end{cases}$$

*is exponent type mean of positive numbers $c$ and $d$ (see [13] (p. 134)).*

**Proof.** Let $g(x,y) = xe^{-(x+y)}, y > 0, x > 0$. From reference [19], we know that function $g(x,y)$ is convex on $\mathbb{R} \times \mathbb{R}$. For $d > 0, c > 0$, and $d \neq c$ by Theorem 2 from $(\frac{d+c}{2}, \frac{d+c}{2}) \prec (d,c)$, it follows that

$$
\begin{aligned}
G(c,d) &= \frac{1}{(c-d)^2} \int_c^d \int_c^d xe^{-x-y} \, dx \, dy \\
&= \frac{1}{(c-d)^2} \int_c^d xe^{-x} \, dx \int_c^d e^{-y} \, dy \\
&= \frac{1}{(c-d)^2} \left(\frac{c+1}{e^c} - \frac{d+1}{e^d}\right) \cdot \left(\frac{1}{e^c} - \frac{1}{e^d}\right) \\
&= \frac{1}{(d-c)^2} \frac{(ce^d - de^c) + (e^d - e^c)}{e^{(c+d)}} \cdot \frac{e^d - e^c}{e^{(c+d)}} \\
&\leq G\left(\frac{d+c}{2}, \frac{d+c}{2}\right) = \frac{c+d}{2} \frac{1}{e^{(d+c)}},
\end{aligned}
$$

which is

$$\frac{ce^d - de^c}{e^d - e^c} + 1 \leq \frac{d+c}{2}e^{(d+c)}\left(\frac{d-c}{e^d - e^c}\right)^2.$$

For the rest, we only need to prove that

$$e^{(c+d)}\left(\frac{d-c}{e^d - e^c}\right)^2 \leq 1. \tag{13}$$

We write $e^d = u$ and $e^c = v$; then, the above inequality is equivalent to the well-known log-geometric mean inequality.

$$L(v,u) = \frac{v-u}{\log v - \log u} \geq \sqrt{vu} = B(v,u).$$

□

**Author Contributions:** Conceptualization, H.-N.S., D.-S.W. and C.-R.F.; Methodology, H.-N.S.; Validation, C.-R.F.; Formal analysis, H.-N.S. and D.-S.W.; Investigation, D.-S.W.; Resources, C.-R.F.; Writing—original draft, D.-S.W.; Funding acquisition, C.-R.F. All authors have read and agreed to the published version of the manuscript.

**Funding:** This research received no external funding.

**Institutional Review Board Statement:** Not applicable.

**Informed Consent Statement:** Not applicable.

**Data Availability Statement:** Not applicable.

**Acknowledgments:** The authors sincerely thanks Chen Dirong and Chen Jihang for their valuable opinions and suggestions.

**Conflicts of Interest:** The authors declare no conflict of interest.

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
