# Peer review of "Schur-Convexity of the Mean of Convex Functions for Two Variables"

_axioms, doi:10.3390/axioms11120681_

Round 1
Reviewer 1 Report
The athours provide a result on Schur-convexity of two variable integral functions relying on Schur's classical result and Hadamard's inequality. Following the main theorem, we can read a few interesting inequalities involving generalized means.
Regarding the presentation of the paper, Lemma 3 is called Leibniz integral rule.
Author Response
Please see the revised version that incorporated the suggestions.
Reviewer 2 Report
See the attached file please.

Author Response

(The authors gave the same response as above.)

Reviewer 3 Report
In this paper the results of Schur convexity established by Elezovic and Pecaric for the average of convex functions have been generalized to the case of the average for two-variable convex functions. As an application, some binary average inequalities have also given.
The paper can be accepted but only after the following revision.
1. The abstract Section needs to be improve, its too short.
2. The structure of the paper should be added
3. The conclusion section is missing. The author should include the conclusion section, where they give some future direction for for further study on this topic.
4. The references must be adjust in one style and in order.
5. The introduction section needs more modification by adding and citing some more recent research on this subject.
6. The paper needs to re-check English-wise, there are some typo and spelling mistakes in the submitted version.
7. The authors should know the correct usage of the commas and full-stops.
Author Response

(The authors gave the same response as above.)

Round 2
Reviewer 2 Report
See the attached file please.

Author Response
Thanks for the comment. We have revised the paper and done some spell checks.
Reviewer 3 Report
The paper is can be accepted but
1. The abstract section needs some modifications
2. The paper is not in the MDPI template
Author Response
In the revised version we made modifications in the abstract. My computer cannot edit MDPi templates though.